# Detection of Aflatoxins B1 in Maize Grains Using Fluorescence Resonance Energy Transfer

**Thanh Binh Nguyen [1], Thi Bich Vu [1,2,\*], Hong Minh Pham [1], Cao Son Tran [3]**,
**Hong Hao Le Thi [3] and Ngoc Thuy Vo Thi [4]**

1 Institute of Physics, Vietnam Academy of Science and Technology, Hanoi 100000, Vietnam;
tbnguyen@iop.vast.ac.vn (T.B.N.); phminh@iop.vast.ac.vn (H.M.P.)

2 Institute of Theoretical and Applied Research, Duy Tan University, Hanoi 100000, Vietnam

3 National Institute for Food Control, Hanoi, Vietnam; sontc@nifc.gov.vn (C.S.T.);
lethihonghao@yahoo.com (H.H.L.T.)

4 Department of Physics, University of Science, Vietnam National University, Ho Chi Minh City 749000,
Vietnam; vtnthuy@hcmus.edu.vn

\* Correspondence: vuthibich@duytan.edu.vn; Tel.: +84-9635-91052

**Abstract:** Aflatoxins are secondary metabolites of *Aspergillus flavus* and *Aspergillus parasiticus*. These fungal species are the most dangerous and common toxin group causing food contamination. Aflatoxin has high toxicity and can cause cancer to humans and animals. The quantitative detection of aflatoxin in food, therefore, plays a very important role. However, in practice, due to low concentrations, aflatoxin detection analysis methods need to be highly sensitive and simple to apply. In this report, the fluorescence resonance energy transfer method (FRET) adopts the donor–acceptor interaction of aflatoxin B1. The CdSe/ZnS quantum dot detection of aflatoxin B1 will be presented wherein the aflatoxin B1 concentration can be determined from the changes in fluorescence lifetime or fluorescence intensity. A fluorescence lifetime calibration curve versus aflatoxin B1 concentrations was established. Test results of aflatoxin B1 determination in maize in Vietnam by FRET method are consistent with the results of aflatoxin B1 determination by HPLC based on ppm concentration.

**Keywords:** Aflatoxin B1 (AFB1); CdSe/ZnS QDs; FRET; fluorescence lifetime; TCSPC; maize

---

## 1. Introduction

Aflatoxins are cancerous secondary metabolites from *Aspergillus flavus* and *Aspergillus parasiticus*. Due to their high toxicity and carcinogenic potential, they are a high concern for the safety of food worldwide. They can be found in various agricultural products such as grains, nuts, spices, etc. Based on chromatographic and fluorescence characteristics, all aflatoxins known to date can be classified into 18 different types, but there are four major naturally occurring aflatoxins: B1, B2, G1, and G2. A third subset, M1 and M2, arise as metabolic products when dairy cattle eat B contaminated grains [1–4]. Among them, aflatoxin B1 (AFB1) is the most common and most widespread [5,6] in the world and accounts for 75% of all aflatoxins contamination of food and feeds [7].

Aflatoxins are widely distributed in nature and are commonly found as contaminants in human food, particularly in cereals, oilseeds, tree nuts, and even milk. They are mutagenic in bacteria and infect cereals during harvest and storage. The host plants are susceptible to *Aspergillus* infection after prolonged exposure in high humidity environments or damaged by prolonged drought conditions. Aflatoxins can be toxic to humans and animals causing liver damage, abnormalities, mutations, and cancer, and when in high doses, aflatoxins can be fatal [8].

Vietnam is a country with a tropical climate which is a favorable condition for the growth of molds such as Aspergillus flavus and A. Parasiticus. There have been great losses for post-harvest and

storage agricultural products, in which the cause of mold accounts for a significant part due to the effects of the tropical climate. When molds grow on grains, the molds not only use the nutrients such as protein, lipid, vitamins, etc., but they also produce toxins that are particularly dangerous to human and animal health [9].

In Vietnam, maize is the second most important staple crop for humans after rice. It is a substitute for rice, especially for people in rural and mountainous areas. It is also the major component of feeds for Vietnam's livestock industry. Unfortunately, maize is also a good substrate for mycotoxins producing fungi, especially those producing aflatoxins. Therefore, from a food safety point of view, it is important to assess and control the levels of aflatoxins in human food and animal feeds [10,11].

There are many methods to detect aflatoxins such as thin-layer chromatography (TLC), liquid chromatography (LC), radioimmunoassay (RIA), enzyme-linked immunosorbent assay (ELISA), immunoaffinity column assay (ICA), and mass spectrometry imaging (MSI) [12]. Conventionally, they often employ expensive equipment requiring trained personnel to operate. A lot of time is also required in preparing the samples besides using large amounts of chemical reagents. The method of using a quick test kit gives low accuracy. Fast, sensitive, and inexpensive detection methods, therefore, have been developed such as fluorescence spectroscopy [13–15] or Fourier transform infrared spectroscopy (FTIR) [16].

In this paper, the detection of AFB1 using fluorescence resonance energy transfer (FRET) method wherein AFB1 acts as a donor, CdSe/ZnS quantum dots (QDs) acts as an acceptor will be presented. The effect of FRET is that the fluorescence intensity and fluorescence lifetime change so that the AFB1 concentration can be determined from changes in fluorescence lifetime or fluorescence intensity. A fluorescence lifetime calibration curve versus AFB1 concentrations was also established. Test results of AFB1 determination in maize in Vietnam by the FRET method are consistent with the results of aflatoxins determination by HPLC in ppM concentration.

## 2. Materials and Methods

### 2.1. Sample Preparation

#### 2.1.1. Semiconductor Quantum Dots

The semiconductor quantum dots (QDs), which were used in FRET measurement, are the CdSe/ZnS shell/core QDs that have a fluorescence emission peak at 535 nm and strong absorption in the ultraviolet region up to 520 nm. CdSe/ZnS QDs were synthesized by the successive ionic layer absorption and reaction (SILAR) method. Details of the preparation conditions have been reported previously [17,18]. QDs particles are spherical in shape with diameters of approximately 4.0 nm. AFB1 and CdSe/ZnS are compatible as donor–acceptor pairs for FRET measurements where CdSe/ZnS QDs concentrations were maintained and the AFB1 concentration were varied as shown in Table 1.

**Table 1.** Concentrations of aflatoxin B1 (AFB1) and CdSe/ZnS for each sample.

| Sample Name | M0 | M1 | M2 | M3 | M4 | M5 | M6 |
|---|---|---|---|---|---|---|---|
| CdSe/ZnS (µL) | 0 | 1000 | 1000 | 1000 | 1000 | 1000 | 1000 |
| Aflatoxin (µL) | 2000 | 1000 | 500 | 100 | 50 | 10 | 0 |
| Methanol | 0 | 0 | 500 | 900 | 950 | 990 | 1000 |

#### 2.1.2. Aflatoxins B1

The reagents for AFB1 used in this study were purchased from Sigma Aldrich with a purity of 98.5% and diluted in methanol solvent at a concentration of 20 µg/mL and then mixed with CdSe/ZnS QDs as shown in Table 1.

Maize seed samples uses directly for human consumption were collected in a Vietnamese market. The samples were dried, analyzed using HPLC at the Vietnam National Institute for Food Control.

The extraction method was based on an official standard method. Twenty-five grams of ground maize seeds, 2 g of sodium chloride, and 125 mL of methanol mixed with water (70:30 ratio volume) were placed in 250 mL conical flask and were blended using a shaker for 2 min. The solid substances were excluded by paper filtration. A 15 mL volume of the solution was then placed into the funnel attached to the IAC column (Vicam Aflatest P 1 mL). The separator column was adjusted at a rate of 1–2 drops per second, and the impurities were washed with 10 mL. The AFB1 was eluted with 1 mL of methanol, and the AFB1 concentration of the maize seeds extract was prepared with different concentrations.

## 2.2. Apparatus

The absorption and steady-state fluorescence spectra of the solutions were recorded using the UV–Vis UV2600 (Shimazu) and the Cary Eclipse (Variant) spectrometers, respectively. Time-resolved fluorescence measurements were performed with a Time-Correlated Single Photon Counting (TCSPC) apparatus. Samples were placed in a 1 cm × 1 cm quartz cuvette with four polished windows and were excited under 405 nm from a picosecond semiconductor laser. The excitation source uses a diode laser with −0.1 mW output power, −10 ps pulse width, and 4 MHz repetition rate, and the detector utilizes a microchannel plate (MCP) photomultiplier tube R3809-50 (Hamamatsu). The average fluorescence lifetimes were estimated by fitting an exponential function to the decay curves.

## 3. Results and Discussion

### 3.1. FRET Theory

In every FRET method, a donor fluorophore (D) initially absorbs the energy due to the excitation of incident light and transfer the excitation energy to a nearby chromophore, the acceptor (A). Energy transfer manifests itself through a decrease or quenching of the donor fluorescence and a reduction of excited state lifetime accompanied also by an increase in acceptor fluorescence intensity [19].

There are four criteria that must be satisfied in order for FRET to occur. These are:

- The fluorescence emission spectrum of the donor molecule must overlap the absorption acceptor molecule.
- Donor and acceptor molecules must be in close proximity to one another (typically 1–10 nm).
- The transition dipole orientations of the donor and acceptor must be approximately parallel to each other.
- The fluorescence lifetime of the donor molecule must be of sufficient duration to allow the FRET to occur.

The non-radiative energy transfer ($k_T$) from donor to acceptor is given by:

$$k_T(r) = \frac{1}{\tau_D}\left(\frac{R_0}{r}\right)^6 \tag{1}$$

where ($r$) is the distance between the donor and acceptor and $R_0$ is the Förster distance, the distance at which the resonance energy transfer is 50% efficient.

The FRET efficiency (E) can be estimated from the Equation (1) using the FRET distance ($R_0$, the distance ($r$) between the donor and acceptor, and the ratio of the acceptor/donor ($n$):

$$E = \frac{nR_0^6}{r^6 + nr^6} \tag{2}$$

The FRET efficiency (E) decreases exponentially by the sixth power of the distance. As the distance between the donor and the acceptor is longer than $2R_0$, the FRET efficiency reaches zero. So, the washing step is not necessary because unhybridized reporter probes are too far from the QDs to affect the FRET signal. The FRET distance ($R_0$) can be estimated from Equation (2) with the quantum

yield (QY) of QDs and the spectral overlap between the emission wavelength of the donor and the absorption wavelength of the acceptor as I as in Equation (3),

$$R_0^6 = \left[ \frac{900(\ln 10)k^2 \varnothing_D}{128\pi^5 N n^4} \right] \int_0^\infty F_D(\lambda)\varepsilon_A(\lambda)\lambda^4 d\lambda \tag{3}$$

where $F_D$ is the fluorescence intensity of the donor without presence acceptor; $\varepsilon_A$ is the extinction coefficient of the acceptor (in $M^{-1} cm^{-1}$); $\lambda$ is the wavelength (in nm); $\Phi_D$ is the fluorescence QDs of the donor in the absence of acceptor; $n$ is the refractive index of the medium; $k^2$ is the orientation factor between donor (D) and acceptor (A) (2/3 for randomly oriented dipoles); and $N$ is the Avogadro's number. The integral part of Equation (3) is known as the spectral overlap integral $J(\lambda)$ and is given by:

$$J(\lambda) = \int_0^\infty F_D(\lambda)\varepsilon_A(\lambda)\lambda^4 d\lambda \tag{4}$$

The energy transfer efficiency can be expressed as:

$$E = \frac{\tau_D k_T}{1 + \tau_D k_T} = 1 - \frac{-F_{DA}}{F_D} = 1 - \frac{DA}{D} \tag{5}$$

where $F_{DA}$, and $\tau_{DA}$ are the relative fluorescence intensity and fluorescence lifetime of the donor in the presence of the acceptor, respectively, and $\tau_D$ is fluorescence lifetime of the donor without the presence of the acceptor.

### 3.2. UV–Vis Absorption and Steady-State Fluorescence Spectroscopy

The UV–Vis absorption and steady-state fluorescence spectra of pure AFB1 and CdSe/ZnS QDs in aqueous solutions are shown in Figure 1. The absorption and fluorescence maxima of AFB1 are centered at 360 and 435 nm, respectively. The absorption and fluorescence maxima are solvent dependent (in this case, the solvent is methanol). The absorption spectrum of CdSe/ZnS QDs has broad absorption that extends from the ultraviolet to the visible region with absorption edge at 520 nm, while the fluorescence spectra show strong emission with maxima at 545 nm.

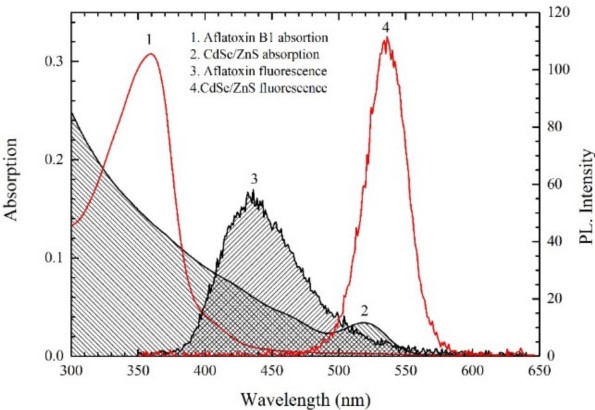

**Figure 1.** The absorption and fluorescence spectra of AFB1 and CdSe/ZnS QDs in aqueous solution. The overlap between AFB1 fluorescence and CdSe/ZnS absorption is shown by cross-line region.

Figure 1 shows that there exists sufficient overlapping of the AFB1 fluorescence spectrum and the CdSe/ZnS QDs absorption spectrum. This is the required condition for FRET between AFB1 and CdSe/ZnS QDs to occur. Here, AFB1 acts as a donor that receives energy from the excitation source and transfers that energy to CdSe/ZnS as an acceptor. The energy transfer results in change of the fluorescence intensity and fluorescence lifetime.

### 3.3. AFB1 and CdSe/ZnS QDs FRET

To study the energy transfer between AFB1 and CdSe/ZnS QDs, a series of AFB1 and CdSe/ZnS QD samples were prepared with the volume ratio as showed in Table 1. Figure 2a and b show the absorption and fluorescence spectra under 405 nm excitation of the AFB1 and CdSe/ZnS QD mixtures.

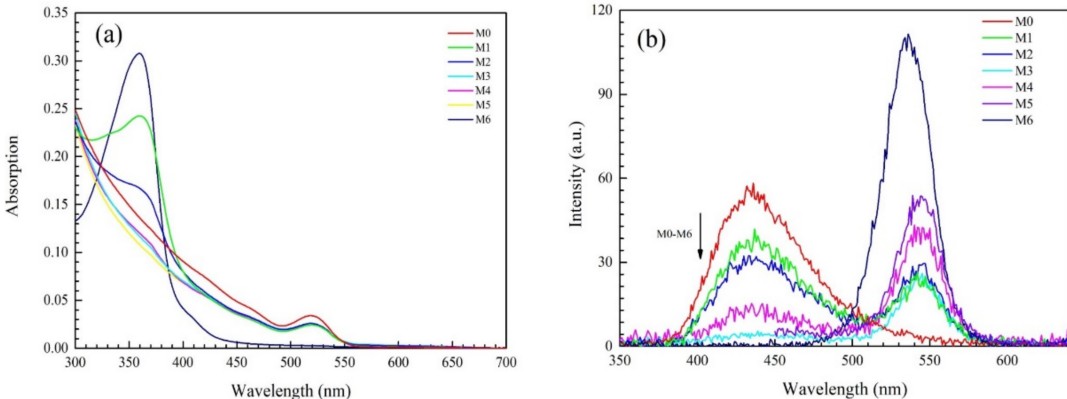

**Figure 2.** (**a**) Absorption spectra and (**b**) fluorescence spectra ($\lambda_{exc.}$ = 405 nm) of AFB1 and CdSe/ZnS QDs mixtures with different volume ratios in methanol.

From Figure 2b, it was observed that the fluorescence spectra of AFB1 and CdSe/ZnS QDs mixtures are characterized by emission peaks at 435 and 545 nm of AFB1 and CdSe/ ZnS, respectively. The fluorescence intensity of CdSe/ZnS QDs without AFB1 (M6) is higher than the fluorescence intensity of pure AFB1 (M0). This may be caused by the excitation wavelength in the weak absorption region of AFB1 and in the strong absorption region of CdSe/ZnS (Figure 2a) aside from the greater quantum yield of CdSe/ZnS QDs compared with AFB1. The fluorescence intensity of AFB1 decreases with respect to pure AFB1 in the mixed samples, while the fluorescence intensity of CdSe/ZnS increases. This is mainly due to the transfer of energy from AFB1 molecule to CdSe/ZnS QDs via FRET. Results show that the absorption peak and emission spectra of AFB1 depends on the solvent solution. The emission peak can also change from 398, 418, 438 nm, etc. depending on the solvents used such as benzene, lipids, or phosphate buffer solution (PBS) [20].

The FRET efficiency can be calculated from the variation of the fluorescence intensity according to Equation (5). However, Figure 2b shows that the fluorescence spectra of AFB1 and CdSe/ZnS QDs are noisy, and the maximum value of the emission peak fluctuates. Moreover, to confirm that the interaction between AFB1 and CdSe/ZnS QDs in solution is FRET, the fluorescence lifetime of the AFB1 and CdSe/ZnS QDs mixtures were performed. Figure 3 shows the time-resolved fluorescence spectra for all samples at room temperature. The spectra have been measured at the peak wavelength of 435 nm, observed in steady-state photoluminescence spectra, under 405 nm excitation. In the case of a sample with only one recombination process (single molecule case), it is possible to fit the fluorescence attenuation path according to the single exponential function. When there are recombinant processes occurring in the sample, it is necessary to fit the curve according to the biexponential, triple exponential, or the average of the stretch-exponent.

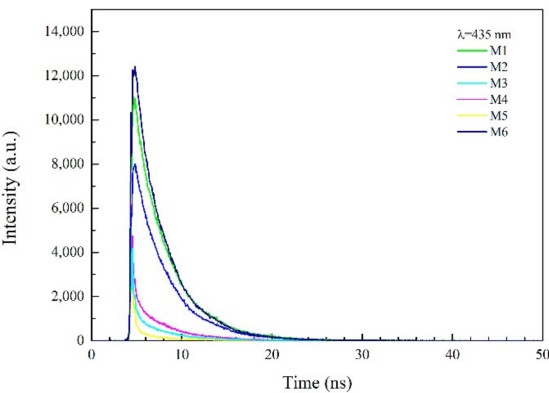

**Figure 3. Figure 3**. Fluorescence decay curves of AFB1 and CdSe/ZnS QDs mixture with the different of the ratio of AFB1 concentration (excited wavelength: 405 nm; emission wavelength (435 nm).

In our case, the decay curves can be well-fitted to a biexponential function described by the equation:

$$I(t) = A_1 \cdot \exp\left(-\frac{t}{\tau_1}\right) + A_2 \cdot \exp\left(-\frac{t}{\tau_2}\right) \tag{6}$$

where $\tau_1$ and $\tau_2$ represent the fluorescence lifetimes for each exponential decay term and $A_1$ and $A_2$ represent their respective amplitudes.

For samples with a high concentration of AFB1 (M6, M1, and M2), the fluorescence decay curves decrease with a single exponential function. On the other hand, for samples with a small concentration of AFB1, their fluorescence intensities are weak, and the fluorescence decay curves decrease quickly and then decrease slowly. Therefore, the fluorescence decay curves were fitted to the biexponential function, where the first fluorescence lifetime (<100 ps) is due to instrumentation response function (IRF), and second fluorescence lifetime is AFB1 fluorescence lifetime in the excited state. The fluorescence lifetime of CdSe/ZnS QDs $\tau_1$ and $\tau_2$ can be attributed to the band edge and the deep trap recombination [21]. The fluorescence lifetime $\tau_1$ almost does not change, while $\tau_2$ increases with increasing AFB1 concentration. Fluorescence lifetime values of AFB1 at 435 nm and CdSe/ZnS QDs at 545 nm were shown in Table 2. Figure 4 shows the variation of AFB1 concentration as a polynomial second-order function of the fluorescence lifetime. This is the result of FRET between AFB1 and CdSe/ZnS QDs which can be used to determine the AFB1 concentration.

**Table 2.** Fluorescence lifetimes of AFB1-CdSe/ZnS samples with different AFB1 concentrations.

| Sample | $\tau_{435}$ | $\tau_{545}$ | AFB1 Concentration (ppM) |
|:---:|:---:|:---:|:---:|
| M0 | 3.76 | - | 2500 |
| M5 | 3.70 | 1.66 | 25 |
| M4 | 3.69 | 1.75 | 125 |
| M3 | 3.57 | 1.80 | 250 |
| M2 | 3.58 | 2.01 | 1250 |
| M1 | 3.50 | 2.17 | 2500 |
| M6 | | 3.81 | - |

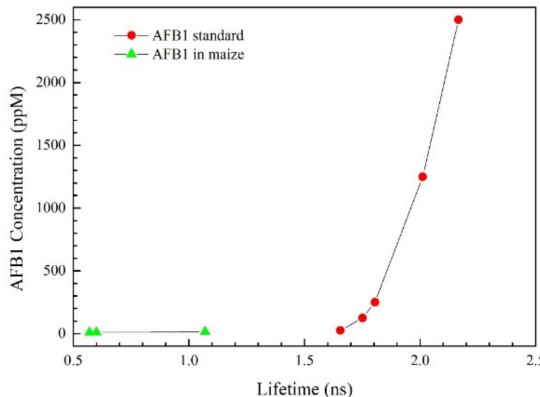

**Figure 4.** Standard reagent AFB1 concentration by fluorescence lifetime.

*3.4. AFB1 Detection*

AFB1 samples with unknown concentrations were extracted from maize by the Vietnam National Institute for Food Controls. AFB1 (S1, S2, S3, and S4) and CdSe/ZnS QDs were mixed with a volume ratio 50:50 in 2 mL methanol. Fluorescence spectra of the mixtures were recorded by Cary Eclipse fluorescence spectrometer. No 435 nm emission peaks was observed, whereas a shoulder at 435 nm as a peak emission of AFB1 in three samples (S1, S2, and S4) besides the emission peak of the CdSe/ZnS QDs have been obtained by a fluorescence spectrometer on the basis of TCSPC under 405 nm excitation. Among four samples, one (S3) does not have this AFB1 emission peak. This is the control sample without AFB1 (Figure 5).

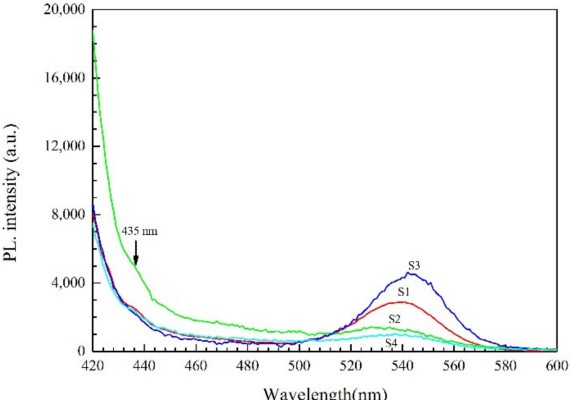

**Figure 5.** Fluorescence spectra of AFB1 extracted from maize–CdSe/ZnS mixture sample. S1, S2, and S4 shows a shoulder at 435 nm chartered AFB1, S3 does not have a shoulder (without AFB1 in sample).

Similar to what was done with the first batch of samples with known AFB1 concentrations, the time-resolved fluorescence spectra for all AFB1 and CdSe/ZnS QDs mixtures have been measured at the peak wavelength of 435 nm under 405 nm excitation. From time-resolved fluorescence decay curves of AFB1 and CdSe/ZnS QDs mixtures (S1, S2, and S4), the fluorescence lifetime of these samples have been determined. These are 1.07, 0.6, and 0.57 ns for samples S1, S2, and S4, respectively. Using the calibration curve in Figure 4, the AFB1 concentrations extrapolated for samples S1, S2, and S4 are 15, 10, and 9 ppM, respectively. These values can be compared with the concentration determined using the HPLC method at Vietnam National Institute for Food Control. Using HPLC method, the AFB1 Concentrations S1, S2, and S4 are 43, 25, and 11 ppM, respectively.

## 4. Conclusions

In summary, a new spectroscopy method for detecting AFB1 concentration based FRET with CdSe/ZnS QDs has been developed. The FRET between AFB1 and CdSe/ZnS QDs causes a change in the fluorescence intensity and fluorescence lifetime. However, fluorescence intensity was weak and has large fluctuations, so the fluorescence lifetimes were used to estimate AFB1 concentration. The results of AFB1 detection in maize showed that the AFB1 concentration detection by FRET was consistent with the results detected by the HPLC method. The application in real sample suggests that this approach shows the promising prospect for the rapid screening of AFB1 in food.

**Author Contributions:** Conceptualization, T.B.N., T.B.V., C.S.T., and H.H.L.T.; data curation, T.B.N., C.S.T., H.H.L.T., and N.T.V.T.; formal analysis, T.B.N., H.M.P., and C.S.T.; investigation, T.B.N., T.B.V., H.M.P., C.S.T., H.H.L.T., and N.T.V.T.; methodology, T.B.N. and T.B.V.; project administration, T.B.V.; resources, T.B.N., C.S.T., H.H.L.T., and N.T.V.T.; validation, C.S.T., H.H.L.T., and N.T.V.T.; writing—original draft, T.B.N. and T.B.V. All authors have read and agreed to the published version of the manuscript.

**Funding:** This work was supported by the Vietnam Academy of Science and Technology under the Program of development in Physics 2020: "Application of time-resolved fluorescence spectroscopy and fluorescence resonance energy transfer (FRET) for assays detection toxin/antibiotics in food" and by the International Center of Physics under the auspices of UNESCO through project number ICP.2019.07.

**Conflicts of Interest:** The authors declare no conflict of interest.

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
