# Peer review of "Detection of Aflatoxins B1 in Maize Grains Using Fluorescence Resonance Energy Transfer"

_applsci, doi:10.3390/app10051578_

Round 1

Reviewer 1 Report

 “Detection of Aflatoxins B1 in Maize Grains Using Fluorescence Resonance Energy Transfer”

 The topic of the research considering the climate of the said country is relevant and very important. But I am very sorry to say that the quality of the work is very poor. The authors have tried to detect Aflatoxins BI using FRET. Here the FRET experiments have been performed erroneously, results are not presented correctly and interpretations are not correct. The authors should follow some standard protocols to design the FRET experiments and then try to present proper explanation. So I should only accept the manuscript after major revision of the experimental technique and manuscript. The sentence construction is not good and so the quality of English should be improved.

The authors perform FRET in methanol. Does Aflatoxin B1 bind inside CdSe/ZnS QDs in this condition and FRET occurs? What is the size of CdSe/ZnS QDs? To perform FRET experiment, the concentration of Donor should be fixed and the concentration of acceptor should be increased to see the changes in the fluorescence signal. Figure 2b: authors say that the decrease in donor fluorescence and increase in acceptor signal is due to FRET. But here the concentration of donor is decreasing and the concentration of acceptor is also increasing. So the results are interpreted wrongly. At the excitation wavelength of Aflatoxin B1, CdSe/ZnS QDs also shows strong absorbance, so use of Aflatoxin B1 and CdSe/ZnS QDs as FRET Pair will be useless. Proper control experiments need to be performed.

Author Response

Dear Professor,

We respectfully submit the revised version of our manuscript entitled "Detection of Aflatoxins B1 in Maize Grains Using Fluorescence Resonance Energy Transfer ".

We would like to thank for providing insight that has helped us revise and improve the paper. We have made the following changes following the your suggestions

Reviewer 2 Report

Authors should correct small mistakes highlighted in the attached text and, above all, adapt the references to the magazine's specifications

Author Response

Dear Professor,

We respectfully submit the revised version of our manuscript entitled "Detection of Aflatoxins B1 in Maize Grains Using Fluorescence Resonance Energy Transfer ".

We would like to thank for providing insight that has helped us revise and improve the paper. 

Reviewer 3 Report

Here you can find a bunch of my thoughts after reading your paper.

English needs improving. The aim of this study should be emphasized more deeply. What is the novelty of your paper? It is not clear for me reading the introduction. Improve the quality of the figures because it is poor. I do not share the authors' opinions that the AFB1 concentration detection by FRET was consistent with the results detected by HPLC method. You missed discussion

Line 16-17

parasiticus and Aspergillus parasiticus ?

Line 34, 45, 50, 179, 191

An ellipsis is typical for fiction literature. We do not use three periods in the scientific literature

Line 41

What do you mean by “they can create bacteria”?

Line 47

Fungi are not aflatoxins

Line 49

You are writing about grains or nuts?

Line 58-60

Either use uppercase or lowercase letters everywhere

Line 86

Are you sure about this unit - µ/g/ml?

Line 88

“Maize extract was prepared from noncontaminated ground maize seeds”. What makes you claim that?

Line 147

are could?

Line 156

Shows

Line 162-165, 216-220, 230-232

Rephrase these sentences

Author Response

Dear Professor,

We respectfully submit the revised version of our manuscript entitled "Detection of Aflatoxins B1 in Maize Grains Using Fluorescence Resonance Energy Transfer ".

We would like to thank for providing insight that has helped us revise and improve the paper. We have made the following changes following the your suggestions:

Round 2

Reviewer 1 Report

All the comments are answered well. 

Reviewer 3 Report

line 40

The phrase “they can create bacteria” means that you take some aflatoxins and they create bacteria like an artist creating a piece of art. Don't you understand how it sounds? Rephrase this.

line 46

Fungi are not aflatoxins. I agree with what you wrote in your explanations (Aflatoxins are a type of mycotoxin produced by Aspergillus species) and I don't understand why do you insist on the phrase “…fungi such as aflatoxins”. Rephrase this.

Author Response

Dear Professor,

We respectfully submit the revised version of our manuscript entitled "Detection of Aflatoxins B1 in Maize Grains Using Fluorescence Resonance Energy Transfer ".

We would like to thank you for your insights that helped us improve our manuscript. We have made the following revisions based on your suggestions:

Line 40

The phrase “they can create bacteria” means that you take some aflatoxins and they create bacteria like an artist creating a piece of art. Don't you understand how it sounds? Rephrase this.

We revised the statement into: “They are mutagenic in bacteria…”

Line 46

Fungi are not aflatoxins. I agree with what you wrote in your explanations (Aflatoxins are a type of mycotoxin produced by Aspergillus species) and I don't understand why do you insist on the phrase “…fungi such as aflatoxins”. Rephrase this.

We revised the statement into: “Vietnam is a country with a tropical climate which is a favorable condition for the growth of molds such as Aspergillus flavus and A. parasiticus.”
